# Prognostic and Predictive Role of Tumor-Infiltrating Lymphocytes (TILs) in Ovarian Cancer

**DOI:** 10.3390/cancers14184344

**Published:** 2022-09-06

**Authors:** Daniele Fanale, Alessandra Dimino, Erika Pedone, Chiara Brando, Lidia Rita Corsini, Clarissa Filorizzo, Alessia Fiorino, Maria Chiara Lisanti, Luigi Magrin, Ugo Randazzo, Tancredi Didier Bazan Russo, Antonio Russo, Viviana Bazan

**Affiliations:** 1Section of Medical Oncology, Department of Surgical, Oncological and Oral Sciences, University of Palermo, 90127 Palermo, Italy; 2Department of Biomedicine, Neuroscience and Advanced Diagnostics, University of Palermo, 90127 Palermo, Italy

**Keywords:** ovarian cancer, prognostic and predictive role, tumor immunology, tumor-infiltrating lymphocytes (TILs), tumor microenvironment

## Abstract

**Simple Summary:**

In this review, we present current knowledge about the prognostic and predictive role of tumor-infiltrating lymphocytes (TILs) in ovarian cancer since this tumor shows a potential immunogenicity which makes it suitable for immunotherapy treatment; although, to date, the immunotherapy has not yielded the expected results. Understanding the factors which drive infiltration could be the key to unravel the clinical outcome heterogeneity in this cancer. Furthermore, understanding molecular mechanisms underlying the crosstalk between cancer and immune cells within the tumor microenvironment (TME) could help to identify and select subsets of OC patients who may benefit from immunotherapy.

**Abstract:**

In the last decade, tumor-infiltrating lymphocytes (TILs) have been recognized as clinically relevant prognostic markers for improved survival, providing the immunological basis for the development of new therapeutic strategies and showing a significant prognostic and predictive role in several malignancies, including ovarian cancer (OC). In fact, many OCs show TILs whose typology and degree of infiltration have been shown to be strongly correlated with prognosis and survival. The OC histological subtype with the higher presence of TILs is the high-grade serous carcinoma (HGSC) followed by the endometrioid subtype, whereas mucinous and clear cell OCs seem to contain a lower percentage of TILs. The abundant presence of TILs in OC suggests an immunogenic potential for this tumor. Despite the high immunogenic potential, OC has been described as a highly immunosuppressive tumor with a high expression of PD1 by TILs. Although further studies are needed to better define their role in prognostic stratification and the therapeutic implication, intraepithelial TILs represent a relevant prognostic factor to take into account in OC. In this review, we will discuss the promising role of TILs as markers which are able to reflect the anticancer immune response, describing their potential capability to predict prognosis and therapy response in OC.

## 1. Introduction

Ovarian cancer (OC) is one of the most common gynecological diseases, ranking as the seventh most common female cancer and the eighth leading cause of mortality among women worldwide, with a five-year relative survival of 49% [1]. Up to 95% of all OCs belong to the epithelial subtype, and around 70% of all epithelial ovarian cancers (EOCs) are represented by the high-grade serous carcinoma (HGSC), followed by clear cell (10%) and endometrioid (10%) carcinoma, low-grade serous carcinoma (<5%), and the mucinous type (3%), whereas only 5% of all OC diagnoses are attributable to tumors originating from nonepithelial parts of the ovary, such as the sex cord, stroma and, rarely, from germ cells [2,3]. The various histological subtypes differ in terms of risk factors, type of dissemination, molecular genetics, response to therapy, prognosis, and survival. About 75% of EOC patients are diagnosed at an advanced stage with a five-year relative survival of 29% [4]. Hereditary predisposition to the disease affects approximately 15–20% of women with EOC, and the identification of pathogenic variants in the *BRCA1* and *BRCA2* genes and other homologous recombination genes is mainly associated with the HGSC subtype [5,6].

Several prognostic factors have been proposed and used to predict the clinical outcome in OC patients. In addition to age, tumor histology, performance status, and volume of residual disease, the presence of tumor-infiltrating lymphocytes (TILs) has been shown to be a positive predictor of survival [7]. TILs are a specific population of T cells with a high tumor-specific immunological reactivity against cancer cells, characterized by specificity, memory, and adaptability [8,9]. In the tumor microenvironment (TME), TILs express an immune receptor called programmed cell death 1 (PD-1), an immunoinhibitory receptor of the CD28 family, which plays an important role in tumor immune escape [10]. PD-1 expression is defined as a hallmark of T cell exhaustion. Indeed, the binding of PD-1 to its receptors, PD-L1 or PD-L2, which are expressed on tumor cells, leads to a reduction in T cell proliferation and the production of inflammatory cytokines, resulting in the inhibition of their ability to target cancer cells [11].

Tumor cells use these inhibitory immune checkpoint-mediated mechanisms to inactivate TILs and bypass their recognition and destruction by the immune system [12]. Since tumor lymphocyte infiltration reflects the tumor-specific immune response, TILs represent a potential marker for assessing the intensity of the anticancer immune response [13].

Antibodies are able to inhibit the PD-1/PD-L1 pathway inducing the activation of T lymphocytes by restoring their anticancer function [14]. T lymphocyte activation requires interaction between the T lymphocyte receptor (TCR) and peptides presented by the major histocompatibility complex (MHC) on antigen-presenting cells (APCs) [12].

Several monoclonal antibodies have been developed and used as immunotherapy for the treatment of various cancers, representing, in recent years, one of the most promising advances in the field of the oncology and demonstrating significant success in clinical practice [15,16,17,18]. Despite their success in other tumor types, the clinical use of checkpoint inhibitors (ICIs) in OC has been disappointing with generally low objective response rates of around 6–15% [19].

In the last decade, the presence of TILs has been recognized as a clinically relevant prognostic marker for improved survival and as an immunological basis for the development of new therapeutic strategies, showing a strong prognostic and predictive value in several malignancies, including OC [13]. In fact, many OCs show TILs whose degree of infiltration is strongly correlated with survival [20]. In particular, the infiltration of CD3+ and CD8+ cytotoxic T cells is associated with a better prognosis in EOC patients [21]. Furthermore, the location of TILs in the tumor has been shown to be prognostically important for OC [22]. In fact, a stronger correlation has been reported when TILs, especially CD8+, are located within the neoplastic epithelium (intraepithelial TILs) than in the stroma (stromal TILs) [23].

A significant inverse correlation was observed between CD8+ T lymphocyte levels and PD-L1 expression on cancer cells, suggesting that PD-L1 suppresses CD8+ antitumor cells [24]. A recent study by Jovanović et al. [25] confirmed an association between TILs and PD-L1 expression on tumor cells, which is more frequent in HGSC. Other studies have positively correlated the association between PD-L1 expression on tumor cells and the presence of TILs, demonstrating a better prognosis in tumors with greater lymphocyte infiltrate [11,26].

Based on this evidence, the aim of this review is to discuss the promising role of TILs as markers which are able to reflect the anticancer immune response, by describing their potential capability to predict prognosis and therapy response in OC.

## 2. Crosstalk between Immune System and Tumor Microenvironment in OC

The TME is the extracellular niche in which the tumor arises and develops, also called “tumor stroma” for solid tumors. TME is predominantly composed of tumor-associated blood vessels, local immune cells, extracellular matrix (ECM), and connective tissue cells [27,28,29,30]. In 1889, Stephen Paget defined the TME as a specific soil for each tumor developing the “seed and soil” theory about metastatic homing [31]. To date, it is clear that the “*TME is not just a silent bystander, but rather an active promoter of cancer progression*” [32]. A close dynamic crosstalk between TME and cancer cells determines a condition of heterogeneity in the TME and tumor itself [33]. In fact, the TME influences tumor features, survival, growth, and metastasis, providing information on how the tumor acts at the level of the host TME, in angiogenesis, and immune resistance (Figure 1). The TME can play a key role in cancer development and affects the clinical outcome of the patient [34,35]. To create a tumor-promoting and immunosuppressive TME, defined as a premetastatic niche (PMN), OC cells release tumor-derived exosomes (TEXs) containing DNA, mRNA, miRNA, proteins, lipids, and other metabolites, which, accumulating in the peritoneal cavity together with immune system cells, prepare the TME for the metastatic dissemination [36,37,38,39,40].The crucial feature of the TME in OC is the ability to be the preferred metastasis site of activated mesothelial cells of the peritoneal cavity, as well as of the omentum adipocytes. On the other hand, the peritoneal fluid enables the transcoelomic spread of ascites-associated cancer cells or multicellular spheroids [41,42]. T cells and macrophages are the most common cell types in OC ascites, together with natural killer (NK) cells, tumor-associated fibroblasts (TAFs), adipocytes, and mesothelial cells [43,44]. Tumors have been shown to be infiltrated with several adaptive and innate immune cells acting in both pro- and anti-tumorigenic roles [35]. TIL local infiltration implies a local immunity of immune cells around or within the tumor [45]. The abundant presence of TILs in OC suggests the immunogenic potential of this tumor. In fact, many tumor-associated antigens (TAAs) can be recognized by T cells, and this underlines the potential benefits of immunotherapy in treatment options for OC [46,47]. TAAs are the target of the immune attack [47]. The sentinel cells of the immune system picking up the expressed TAAs on the surface of tumor cells are the dendritic cells, which bring the cancer proteins to the lymph nodes and activate the T cell-mediated immunity, making the T cells able to evade the working station of the immune system and attack the tumor [48]. These physiological events happen in all OC patients with better survival, as the naturally generated lymphocytes are too few to attack and destroy the tumor beforehand. However, this mechanism can be boosted by tumor vaccines, and immunotherapies can improve the activity of the dendritic cells and lymphocytes [49,50]. Several cells of the immune system in TME represent a critical component for OC progression [51,52,53]. The tumor-associated macrophages (TAMs) represent the main type of immune cells identified in the TME of OC [54]. TAMs are plastic cells and can be polarized by colony-stimulating factor-1 released by tumor cells into an immunosuppressive M2-like phenotype, producing cytokines, chemokines, enzymes, and exosomes having miRNA, by promoting tumor metastasis. They are classified into two major phenotypes, such as M1 and M2, with different and complex roles. M1 TAMs have a suppressive function on tumors, whereas M2 TAMs enhance cancer progression [41,55,56,57]. Moreover, TAMs interact with other lymphocytes, NK cells, and dendritic cells, favoring immunosuppression [54,58]. The cancer immune surveillance in TME is in opposition to the immune-resistant mechanism, namely, immune evasion [46,59,60]. The evasion is induced by several mechanisms, such as the reduction and/or loss of TAA expression, down-regulation of MHC, overexpression of the antiapoptotic effector B-cell lymphoma 2 (BCL-2), increasing the amount of immunosuppressive regulatory cells (Tregs, MDSCs), and the expression of inhibitory molecules, including CTLA-4, PD-L1, and Fas ligand [46,61]. Despite the high immunogenic potential, OC has been described as highly immunosuppressive with a high expression of PD-1 by TILs [20,62]. Numerous findings led to the knowledge that in OC patients, a great number of TILs and other activated immune stimulatory mechanisms are positively associated with increased PFS and OS, while immune evasion events cause a worse prognosis [24,63,64,65]. Therefore, immunotherapies, which are aimed to enhance the host immune response and/or reduce immune evasion, assume a crucial role in the fight against OC [66]. Several cells are involved in immunosuppression of OC, such as TILs, Tregs, dendritic cells, myeloid-derived suppressor cells (MDSCs), and macrophages [67,68]. Curiel et al. [69] showed the association between Treg-mediated pathogenesis and poor prognosis in OC [70,71]. The immunosuppressive Treg lymphocytes are enrolled in tissue by a preferential migration induced by an up-regulation of the chemokine CCL22, and they seem to be the major factor in inhibiting the immune system-mediated fight against OC [68]. An amount of cytokines and growth factors, produced and released into TME by cancer cells, immune system cells and other host cells allow tumor progression and metastasis escape [41].

The main stimulated mechanisms are: (i) proliferation, growth, and survival of neoplastic cells (EGF family, IL-6, TGFβ) [72,73,74]; (ii) epithelial mesenchymal transition (EMT) and metastasis (TGFβ) [74,75]; (iii) angiogenesis (VEGF, FGF, CXCL8/IL-8) [76,77]; (iv) migration of immune system cells in the tumor site (CCL family, CXCL family) [78]; (v) immunosuppression (VEGF, IL-6, IL-10, TGFβ) [79]; (vi) stemness (KIT ligand, R-spondins) [80,81]; and (vi) ascitic fluid formation (VEGF) [82,83].

To date, the crosstalk between TME and tumor cells is regulated to a great extent by molecular factors, and the related signaling pathways are still unclear.

## 3. TILs in Ovarian Cancer

TILs are white blood cells which migrate from blood circulation into the tumor, participating in the constitution of TME. They can be found in the stroma and within the tumor and include CD8+, CD4+ T cells, and CD20+ B cells [84,85]. TILs are represented not only by T and B cells but also by natural killer (NK) lymphocytes [7,84]. The main subgroups of T lymphocytes are the so-called “Helper T-Cells” (CD4+), “Cytotoxic T-Cells” (CD8+), and “Regulatory T-Cells” (CD4+, CD25+ and FOXP3+), playing a role in controlling tumor growth [7,86,87].

The thymus is an organ of relevance in which T cell development and differentiation occurs and is characterized by two main subsets of cells: medullary and cortical thymic epithelial cells, which play an important role in the selection of T cells [88].

Interestingly, an association was found between CD8+ T cells and longer survival in various tumors, and the presence of TILs is considered a relevant prognostic factor, for example, in colorectal tumor and melanoma [89,90,91]. Moreover, the role of TILs (in particular, CD4+ and CD8+ T lymphocytes, and CD20+ B-Cells) was also established in OC [92].

Regulatory T cells have an inhibitory activity, and their presence inside TME seems to be correlated with a poor prognosis in OC [71,93]. The OC histological subtype which has the major abundance of lymphocyte infiltrates (CD20+, CD25+, FoxP3+) is HGSC, followed by the endometrioid type, whilst mucinous and clear cell ovarian tumors seem to express a lower percentage of TILs [84].

Intraepithelial CD8+ TILs and high CD8+/Treg ratio are associated with favorable prognosis compared to stroma TILs in OC [70].

TILs showed prognostic and predictive value in different solid tumors, including OC [64]. A meta-analysis including 1815 OC patients demonstrated that a lack of intraepithelial TILs (CD3+ or CD8+) was associated with worse survival [65]. Moreover, a further analysis of immune infiltrates from 199 optimally debulked patients with HGSC also showed that TIA-1 (marker of CD8+ cytotoxic T cells), FoxP3 (regulatory T cells, Treg marker) and CD20 (marker of B cells) are favorable prognostic factors [92]. Further studies showed that CD20+ B cells co-localized with CD8+ T cells in HGSCs, increasing patient survival compared to CD8+ TILs alone [94]. In particular, both T cells and B cells are able to form lymphoid structures in HGSC, and often B cells express plasma cell features, determining a prognostic benefit [95]. In contrast with the previous reports, other studies showed that CD4+CD25+FoxP3+ regulatory T cells were associated with worse prognosis in OC [69,96].

Immune checkpoint inhibitors targeting PD-L1 or PD-1 could be a potential treatment for OC, as infiltration of intraepithelial CD3+ and CD8+ TILs is correlated with survival [20,24,97].

The PD-1 and PD-L1 protein expression, assessed in cancer cells and TILs, respectively, from 215 primary HGSCs and CD3 expression in TILs have been also shown to be positive prognostic factors for survival [98].

In addition CD103, a subunit of alpha E/beta7 integrin, which binds to E-cadherin on epithelial cells, was found coexpressed in TILs from primary OCs (HGSC, endometrioid, mucinous, and clear cell) [99]. Alpha E/beta7 integrin is expressed by 2% of circulating lymphocytes and by 90% of intraepithelial lymphocytes, especially by some CD4+ and CD8+ cells [100]. The alpha E/beta7 integrin facilitates the retention of effector and memory lymphocytes in the gut epithelium through interaction with E-cadherin [100]. CD103+ TILs are the most represented in HGSC and are preferentially localized in the epithelial regions of tumors. CD103 was predominantly present in CD8+ T cells expressing activation (HLA-DR, Ki-67, PD-1), cytolytic (TIA-1) markers, and in CD56+ NK cells [99]. The presence of CD103+ TILs has been associated with the survival of the patient in HGSC by different research groups [99,101].

An increased density of TILs has been observed after neoadjuvant chemotherapy in HGSC, although TILs-negative tumors generally remained negative [102].

Recently, Dai et al. [103] have proposed the use of nomograms as clinically easy-to-implement and reliable calculation models for predicting the density of TILs in the TME of HGSOC women in order to monitor the immune status of patients and adopt suitable treatment options. Through a correlation analysis between TILs, clinical features, and blood indicators (inflammatory and tumor markers), the authors showed that age, menopausal status, estrogen receptor (ER), Ki-67 index, platelets, white blood cell (WBC) count, serum carbohydrate antigen 153 (CA153) levels, and lactate dehydrogenase (LDH) were associated in a linear way with the density of CD3+, CD8+, or CD4+ TILs in the TME [103].

As far as genetic diseases are concerned, in one study on endometrioid OC which is associated with Lynch syndrome, TILs and peritumoral lymphocytes did not show any relationship with mismatch repair (MMR) status/microsatellite instability (MSI) [104,105]. TILs are elevated in *BRCA1*-mutated ovarian carcinomas and have been included in a set of histological criteria predicting *BRCA1* mutations in HGSCs, although showing only a high negative predictive value [106]. No clear relationship has been found between levels of TILs and *BRCA2*-mutated OCs [107]. However, another study involving *BRCA1/2*-mutated HGSCs showed a higher mutational load and increased TILs and PD-1/PD-L1 expression compared to tumors without alterations in homologous recombination (HR) genes [108]. Moreover, *BRCA1/2*- and *TP53*-mutated OCs showed higher levels of PD-1 and PD-L1 [109].

Finally, several studies were carried out to identify different molecular subtypes of OC based on different molecular and immunological characteristics of tumor. In particular, some researchers performed investigations which allowed the classification of HGSC into four different subtypes, called C1 (mesenchymal), C2 (immunoreactive), C4 (differentiated), and C5 (proliferative) [110]. Chen and colleagues [111], instead, identified three OC subtypes (C1, C2, and C3) which showed differences in immune score with significant impact on prognosis and therapy response. In particular, an increased quantity of B cells, CD8+ T cells, NK cells, cytotoxic lymphocytes, neutrophils, monocytes, Th2 cells, effective memory T cells (Tem), and Treg cells was observed in the C1 subtype [111]. Additionally, Liu et al. [112] performed combination analyses between PD-1/PD-L1 and TILs in order to characterize a unique molecular subtype of HGSC with higher APC infiltration and greater PD-1/PD-L1 expression which is able to benefit from a specific immunotherapy.

## 4. Prognostic Role of TILs in Ovarian Cancer

Until now, many attempts have been made to clarify the role of TILs and other immune cells within the tumor microenvironment in several solid tumors [8]. The role of TILs as prognostic biomarkers in patients with OC, one of most aggressive among gynecological cancers, has been recently explored. The correlation between the exact location, density, subpopulation of TILs within the tumor mass, and clinical outcome is still partially defined [63]. Over the last two decades, the awareness on this topic is growing, leading to several studies. According to some authors, a high concentration of intraepithelial CD3+, CD8+, and CD103+ cells seems to be correlated with a better clinical outcome [64,70,101,113].

Moreover, the presence of intratumoral CD3+ cells was associated with reduced recurrence rate and increased overall survival (OS) in advanced OC patients after surgery and adjuvant chemotherapy [64,69,114]. Conversely, the studies by Sato et al. [70] and Clarke et al. [23] suggested that CD3+ TILs did not seem to influence patient prognosis.

Low intraepithelial CD8+ TILs appear as predictors of worse prognosis in advanced OC patients, while the presence or absence of CD8+ cells within the stroma had no statistically significant influence on the clinical outcome [22,23,24,69,70,115,116,117].

High grade serous OC (HGSOC) has been shown to be more frequently associated with increased levels of CD8+ TILs, and this was positively correlated with OS, not only in HGSOC but also in endometrioid carcinoma [92,118].

Additionally, higher CD8+/CD4+ TIL and CD8+/Treg+ TIL ratios were correlated with a better prognosis than low ratios in 117 patients, with a median survival of 74 versus 25 months and 58 versus 23 months, respectively [70]. According to previous data, patients whose primary or metastatic lesions contained a high density of CD8+ TILs and high CD8+/FOXP3+ cells ratio had an increased disease specific survival (DSS) [68,115].

The positive association between intraepithelial CD3+ and CD8+ TILs and survival in OC patients was observed, in 2012, in a meta-analysis by Hwang et al. [65], including 10 studies and 1815 OC patients. This was confirmed more recently by the meta-analysis of Li et al. [97], which included 21 different studies and almost 3000 patients.

There is conflicting data regarding Treg cells, the majority of which sustain the correlation with a poor prognosis [69,70], while other studies found no association or even a positive effect on OS [92,115].

In addition, high levels of CD20+ TILs appear to be correlated with improved survival in OC patients [92,116]. Furthermore, the presence of B cells and NK cells in ascites and pleural effusion could predict a worse clinical outcome [119].

The prognostic benefit associated with increased TILs is unclear in women who underwent a neoadjuvant chemotherapy (NACT) [120,121], although a retrospective study involving 131 patients showed a potential prognostic role for TILs after NACT in patients with EOCs, suggesting a reduced mortality risk associated with an increased presence of TILs [122].

Additionally, PD-1 and PD-L1 can be considered independent prognostic factors in OC [24,98].

A recent work, in accordance with a previous study [123], showed a negative correlation among high levels of TILs CD8+ T cells and PD-L1 expression in HGSOC, confirming that tumor PD-L1 expression is a predictor of TIL deficiency in OC [124]. However, as previously reported [11], the combination of PD-L1 expression with CD8+ TILs positively correlates with survival more than CD8+TILs alone, suggesting a potential prognostic role of the PD-L1/CD8+ TILs combination [124].

### TILs and Homologous Recombination Deficiency (HRD)

An increased number of TILs, stimulated by tumor-specific neoantigens, is recruited in hypermutated solid tumors [125].

According to previous reports on solid tumors, the presence of the *BRCA1* or *BRCA2* pathogenic variant, leading to a high mutational load and neoantigens production, was associated with the increased presence of immune infiltrates [126]. In fact, patients with HR-deficient OC, who showed abundant CD3+ and CD8+ cells, experienced an improved clinical outcome compared to HGOC women without mutations (HR-proficient OC) [108,127]. The number of CD8+ TILs and their link with better outcome differ on the basis of the mutational status, with a favorable association presented by *BRCA1* mutation carriers compared to *BRCA2*-mutated or *BRCA1/BRCA2*-*wild-type* patients [23,118,128].

Finally, *BRCA1/2*-mutated OCs also exhibited a significantly increased expression of immune checkpoint modulators, such as PD-1 and PD-L1, suggesting an interplay between HR status, the immune microenvironment within the tumor, and the clinical outcome of HGOC patients [108].

## 5. Predictive Role of TILs in Ovarian Cancer

Another important aspect is represented by the modulatory role of antineoplastic drugs on TILs. This is a very interesting field, full of continuous scientific discoveries still being defined [20,129,130]. In fact, following the demonstration of the importance of subpopulations of TILs and the discover of the immunogenicity in ovarian tumors, multiple therapeutic strategies have been specifically studied and developed [120,131,132]. Various chemotherapy drugs can modulate the activity of distinct immune cell subsets or the immune phenotype of tumor cells, by enhancing antigen presentation, increasing the expression of costimulatory molecules, down-regulating immune checkpoint molecules, or promoting tumor cell death [132]. Conventional cytotoxic drugs may harness the antineoplastic immune microenvironment capacity through a number of mechanisms, including the induction of cellular rearrangements that make the tumor cell more attackable by the host immune surveillance system or by inducing cell death [120]. With regards to the antineoplastic drugs commonly used as neoadjuvant treatment for OC, there is increasing data to support that both carboplatin and paclitaxel may rearrange tumor-mediated immunosuppressive mechanisms and consequently cause effects on immune response [133,134]. Furthermore, carboplatin plus paclitaxel chemotherapy, beyond killing tumor cells, has been also shown to reduce immunosuppression for 2 weeks coinciding with an enhanced T cell immunity, by potentially creating a window of opportunity to identify the optimal time to introduce an ICI-based therapy [120,135,136,137]. In particular, ICIs targeting cytotoxic T lymphocyte-associated protein 4 (CTLA-4) or PD-L1/PD-1 are among the most innovative molecules in this field [138]. Anti-CTLA-4 antibodies improve naive T cell priming and activation in lymph nodes, where they eventually move to promote tumor rejection. The combination of the PD-L1/PD-1 and CTLA-4 blocking may be meaningful in OC, according to the emerging evidence [139].

The combination of ICIs and poly (ADP-ribose) polymerase inhibitors (PARPs) is a very interesting and promising aspect. An enzyme called PARP is in charge of fixing DNA single-strand breaks. The immune system can recognize neoantigens that accumulate as a result of its suppression. Additionally, a less effective DNA repair system leads to an accumulation of cytosolic DNA, which, in turn, triggers the production of interferons and chemoattractants in the proinflammatory cGAS-STING pathway, increasing the response of the immune system against cancer cells [140].

Angiogenesis and immune suppression are two important related biological processes [141]. Both mechanisms promote tumor growth and immunological tolerance. To date, several studies have focused on angiogenic tumor endothelium as a passive physical barrier that inhibits T cell extravasation and effectively counteracts tumor immunity through anergy [142,143,144]. The recent data indicate that the tumor endothelium also acts as an active immunological regulator capable of directly inhibiting the function of T lymphocytes [141,145]. An immunosuppressive environment is specifically promoted by angiogenic growth factors by inducing the expression of FasL on the tumor endothelium, which preferentially kills tumor-reactive CD8+ cells [141]. The existing research suggests that angiogenesis could mechanically inhibit lymphocyte T migration and endothelial adhesion [141,146] because patients who typically have enhanced tumor angiogenesis do not have TILs [64]. Many researchers found that angiogenesis inhibition during adoptive therapy increases the infiltration of anticancer T lymphocytes [142,144,147]. In particular, the inhibition of tumor angiogenesis predominantly encourages the infiltration of effector CD8+ T cells by preventing the effector T cell death caused by the FasL expression on the tumor endothelium [141]. Vascular endothelial growth factor (VEGF) plays a key role in healthy and pathological angiogenesis and is a key factor in tumor development [20,148]. In the OC microenvironment, VEGF has been shown to be highly expressed [149]. It allows the vascular permeability, tumor angiogenesis and spread of peritoneal OC through the development of malignant ascites [150]. VEGF not only promotes tumor angiogenesis but also has immunosuppressive properties. Consequentially, the activity of T lymphocytes is inhibited by VEGF, which also helps induce and maintain regulatory T lymphocytes (Treg), prevent the functional maturation of dendritic cells, increase the expression of inhibitory immune checkpoints on CD8+ lymphocytes, and promote the presence of macrophages associated with cancer [20]. Antiangiogenic drugs could reverse the immunosuppression caused by VEGF, improving the efficacy of ICIs in OC. In vitro inhibition of VEGF has been shown to increase the activation of cytotoxic T lymphocytes and decrease the expression of the inhibitory factors of T lymphocytes, such as PD-L1, TIM-3, LAG-3, and CTLA-4 [151]. Currently, many studies are underway with combined therapies between immunotherapies and antiangiogenics. Although immunotherapy or antiangiogenic therapy have often been used as monotherapies, in the light of this new evidence, the close relationship between angiogenesis and immunosuppression is very clear [141].

Ongoing and completed main studies concerning the modulating action of therapies on TILs are reported in Table 1.

## 6. Conclusions and Future Perspectives

The tumor microenvironment plays a key role in cancer development and the clinical outcome of OC patients. Dynamic interactions between tumor cells and their surrounding microenvironment influence tumor survival, growth, and metastasis. Immune system cells can offer a critical checkpoint for tumor progression. However, the infiltration of the tumor by immune cells appears to have a dual antithetic function: prometastatic or antimetastatic. Tumor infiltration by TILs has emerged as a good prognostic marker in numerous tumors, including OC. However, the main molecular factors regulating the crosstalk between tumor cells and T lymphocytes and the related signaling pathways are still unclear.

Although further studies are needed to better define their role in prognostic stratification and the therapeutic implication, intraepithelial TILs represent a considerable prognostic factor to take into account in OC. In fact, TILs as prognostic biomarkers have some clear advantages above others, such as the ease, speed, and low cost with which the analysis can be performed and the high reproducibility of the scoring. Conversely, the assessment of genomic biomarkers and PD-L1 are expensive, laborious, and not always easily implemented in low-to-middle income countries.

This review confirms the importance of TILs as a precious addition to the set of standard prognostic factors in OC patients. In the future, understanding the factors that drive infiltration will be the key to unravel the clinical outcome heterogeneity in this cancer.

Several experimental studies showed that TILs should be considered for prospective clinical trials investigating (neo)adjuvant chemotherapy de-escalation strategies. Furthermore, the absence of TILs could predict the failure of the PD-L1 blockade. The checkpoint blockade after infusion of TILs has been shown to be a potentially promising approach to increase response rates and induce long-term survival. Understanding molecular mechanisms underlying the crosstalk between cancer and immune cells within the TME will help to identify and select subsets of OC patients who may benefit from the immunotherapies.

## Figures and Tables

**Figure 1 cancers-14-04344-f001:**
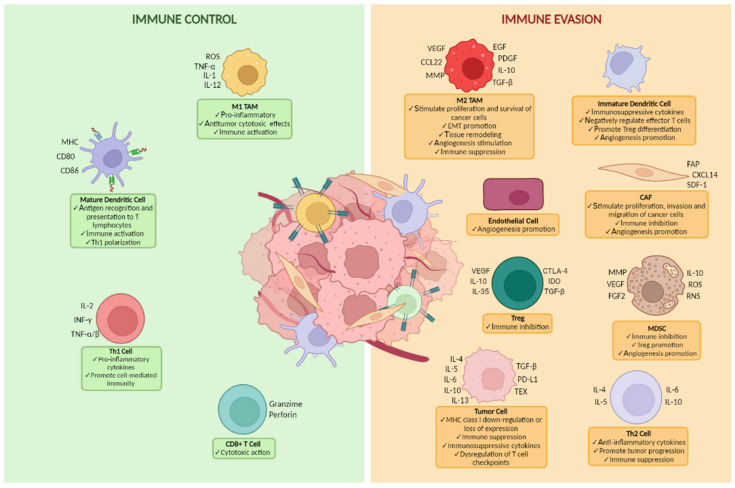
Cellular components and functions of the tumor microenvironment in ovarian cancer. The image was created with BioRender Software (https://biorender.com/ (accessed on 11 July 2022)). Abbreviations: TAM, tumor-associated macrophages; EMT, epithelial mesenchymal transition; CAF, cancer-associated fibroblast; MDSC, myeloid-derived suppressor cell.

**Table 1 cancers-14-04344-t001:** Ongoing and completed main studies concerning the modulating action of therapies on TILs.

Study	Trial Identifier	Study Description	Phase	Status	Number of Patients
**Adoptive T Cell Therapy in Patients with Recurrent Ovarian Cancer (OVACURE)**	**NCT04072263**	Ovarian cancer is a highly immunogenic tumor, and good survival is tightly linked to the presence of tumor-infiltrating CD8+ T cells and the absence of immunosuppressive immune cells. This clear correlation between T cell infiltration and disease progression suggests that OC may be sensitive to adoptive cell therapy by infusion of ex vivo expanded autologous TILs provided that immune suppression is reduced.	1–2	Recruiting	12
**T Cell Therapy in Combination with Nivolumab, Relatlimab, and Ipilimumab for Patients with Metastatic Ovarian Cancer**	**NCT04611126**	In two consecutive pilot trials, adoptive cell therapy (ACT) with TILs was combined with a CTLA-4 inhibitor, Ipilimumab, and a PD1-inhibitor, Nivolumab. About 90–100% of infused T cells expressed LAG-3. The interaction between LAG-3 on T cells and MHC-II on tumor cells inhibits T cell function. In this study, adding the LAG-3 antibody, Relatlimab, to the ACT regimen described above may therefore unleash T cell antitumor efficacy by blocking the known LAG-3–MHC-II interaction.	1–2	Recruiting	18
**TIL Therapy in Combination with Checkpoint Inhibitors for Metastatic Ovarian Cancer**	**NCT03287674**	Before TIL infusion, the patients receive 1 week of preconditioning chemotherapy with cyclophosphamide and fludarabine. After TIL infusion, Interleukin-2 is administered to support T cell activation and proliferation in vivo. Mainly transient clinical responses were observed, and therefore, the investigators plan to combine TIL therapy with checkpoint inhibitors to potentially increase the clinical effect.	1–2	Completed	7
**The ACTIVATE (Adoptive Cell Therapy InVigorated to Augment Tumor Eradication) Trial**	**NCT03158935**	Patients will first receive either cyclophosphamide or cyclophosphamide and fludarabine. These are chemotherapy agents that prepare the body to receive TILs. Patients are then infused with autologous TILs. Following TILs infusion, patients will receive low-dose IL-2 therapy that is intended to activate and stimulate the growth of cells in the immune system of the patients. If the patients meet the required criteria, they will be given pembrolizumab.	1	Completed	8
**Immunotherapy Using Tumor Infiltrating Lymphocytes for Patients with Metastatic Cancer**	**NCT01174121**	The purpose of this study is to see if these specifically selected tumor-fighting cells can cause digestive tract, urothelial, breast, or ovarian/endometrial tumors to shrink and to see if this treatment is safe.	2	Recruiting	332
**“Re-Stimulated” Tumor-Infiltrating Lymphocytes And Low-Dose Interleukin-2 Therapy in Patients with Platinum Resistant High Grade Serous Ovarian, Fallopian Tube, or Primary Peritoneal Cancer**	**NCT01883297**	Prior to the main treatment, patients will receive cyclophosphamide by vein. Patients will then receive an infusion by vein of autologous TILs, stimulated with certain substances called autologous dendritic cells (DCs) and OKT3 (anti-CD3 antibody) given to the patients as an infusion. After infusion of TILs, low-dose interleukin-2 (IL-2) therapy will be given.	1	Active, not Recruiting	3

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
