# Peer review of "Prognostic and Predictive Role of Tumor-Infiltrating Lymphocytes (TILs) in Ovarian Cancer"

_cancers, 2022, doi:10.3390/cancers14184344_

Round 1
Reviewer 1 Report
This review is an advancement on the previous review on 'Tumor Infiltrating Lymphocytes in Ovarian Cancer' by Santoiemma, P.P. ; Powell, D.J. in 2015.
Numerous findings have demonstrated that TIL are relevant prognostic markers for improved survival in OC. This work is commendable and contributes to the development of immuno-therapies to improve the survival outcomes of OC patients. There are some suggestions: which include the inclusion of two very recent works in this field;
Nomograms to Predict the Density of Tumor-Infiltrating Lymphocytes in Patients With High-Grade Serous Ovarian Cancer. Front. Oncol., 25 February 2021 Sec. Molecular and Cellular Oncology. https://doi.org/10.3389/fonc.2021.590414 and S. Giovannoni, A. Garbi, G. Parma, M. Lapresa, E. Zaccarelli, A. Vingiani, I. Ardoino, G. Pruneri, N. Colombo - Tumour-infiltrating lymphocytes (TILs) in patients with epithelial ovarian cancer undergoing neoadjuvant chemotherapy: A retrospective study, Annals of Oncology, Volume 30, Supplement 5, 2019, Page v416, https://doi.org/10.1093/annonc/mdz250.026. I recommend the inclusion of these articles which further augments your findings. The work requires some grammatical corrections. Also reference 45 requires a slight correction.
Author Response
We are grateful to you for your accurate revision of our manuscript and your helpful suggestions.
1) As you suggested, we cited and discussed two recent works by Dai et al. (Section 3 “TILs in Ovarian Cancer”; reference #103) and Giovannoni et al. (Section 4 “Prognostic role of TILs in Ovarian Cancer”; reference #122), respectively.
2) We have corrected some grammatical error and reference #45 in the Bibliography section.
Reviewer 2 Report
The topic of this article is related to dramatically important field concerning the increase of ovarian cancer treatment strategies. Although the shifting of pathogenesis paradigm the effective treatment for the most aggressive ovarian cancer, HGSC has yet to be investigated. Immunotherapy demonstrated inspiring results for some cancers including urothelial carcinoma, melanoma, cervical cancer but the ovarian cancer treatment with such drugs was disappointing. Only moderate effect was shown for the vast majority of ovarian cancer patients. And in this article the authors try to explain the mechanisms of such ambiguous response to therapy. Unfortunately, the authors do not cover molecular subtypes of HGSC because these cancers can demonstrate different clinical, histological and molecular characteristics as well as different prognosis and immunotherapy response. TILs characteristics also differ in mesenchymal, proliferative, differentiated and proliferative molecular subtypes. I suppose that this information should be presented in the paper. As a source of information I recommend such articles as Combined PD-1/PD-L1 and tumor-infiltrating immune cells redefined a unique molecular subtype of high-grade serous ovarian carcinoma. BMC Genomics 23, 51 (2022) and Chen Z, Jiang W, Li Z, Zong Y, Deng G. Immune-and Metabolism-Associated Molecular Classification of Ovarian Cancer. Front Oncol. 2022 May 12;12:877369
Author Response
We thank you for the time you have devoted to the revision and your helpful suggestions.
1) As you suggested, we cited and discussed two recent works by Chen et al. (Section 3 “TILs in Ovarian Cancer”; reference #111) and Liu et al. (Section 3 “TILs in Ovarian Cancer”; reference #112), respectively. In addition, we cited and discussed also the paper by Leong and colleagues (references #110) concerning the molecular subtype classification of high-grade serous ovarian cancer.